# Analysis of Crop Consumption Using Scatological Samples from the Red-Crowned Crane *Grus japonensis* in Eastern Hokkaido, Japan

**DOI:** 10.3390/ani13203167

**Published:** 2023-10-10

**Authors:** Ayaka Yokokawa, Wenjing Dong, Kunikazu Momose, Hiroko Iima, Tomoo Yoshino, Kenichi Izumi, Yusuke Kawai, Tomoko Amano, Tatsuro Nakamura, Akira Sawada, Daiji Endoh, Nobuyoshi Nakajima, Hiroki Teraoka

**Affiliations:** 1School of Veterinary Medicine, Rakuno Gakuen University, Ebetsu 069-8501, Japan; yokokawa-ayaka@bozo.co.jp (A.Y.); dongwenjing_2015@163.com (W.D.); t-naka@rakuno.ac.jp (T.N.); dendoh@rakuno.ac.jp (D.E.); 2NPO Red-Crowned Crane Conservancy, 9-21 Wakatake-Cho, Kushiro 085-0036, Japan; dzi00244@nifty.com; 3Kushiro Zoo, 11 Shimoninishibetsu, Kushiro 085-0204, Japan; hiroko.iima@city.kushiro.lg.jp (H.I.); garrulaxcanorus@yahoo.co.jp (T.Y.); 4College of Agriculture, Food and Environment Sciences, Rakuno Gakuen University, Ebetsu 069-8501, Japan; izmken@rakuno.ac.jp (K.I.); amano@rakuno.ac.jp (T.A.); 5Laboratory of Toxicology, Department of Veterinary Medicine, Obihiro University of Agriculture and Veterinary Medicine, 2-11 Inada-cho Nishi, Obihiro 080-8555, Japan; ykawai@obihiro.ac.jp; 6Biodiversity Division, National Institute for Environmental Studies, Tsukuba 305-8506, Japan; akira.sawada.1312@gmail.com (A.S.); naka-320@nies.go.jp (N.N.)

**Keywords:** amplicon sequencing, crop consumption, *Grus japonensis*, Japan, scatological samples

## Abstract

**Simple Summary:**

The red-crowned crane (*Grus japonensis*), which is an endangered and highly protected bird species, is distributed in two populations: a mainland population in far eastern Eurasia and an island population in Hokkaido, Japan. Red-crowned cranes in Japan are resident birds mainly in the eastern part of Hokkaido. As omnivores, they feed on plants, grains, insects, and fish. Most cranes spend the winter around feeding stations in southeastern Hokkaido, where people provide corn. Since most of the cranes in Hokkaido now live near areas inhabited by humans, cases of crop damage caused by cranes have recently been reported. This study showed that the cranes feed on various crops of human origin, mostly outside farmlands.

**Abstract:**

Total DNA extracts from the intestinal contents of 60 flying red-crowned cranes (juveniles, subadults and adults) found dead in 2006–2021, and the feces of 25 chicks collected in June and July of 2016–2018, were used for PCR reactions with primers specific for 16 crops, followed by high-throughput sequencing. The most predominant crop detected was corn in adult and subadult cranes (61.7%). Other grains (barley, wheat, soybean) (5.0–8.3%) and vegetables (tomatoes, Chinese cabbage, etc.) (1.7–6.7%) were also detected in flying cranes. Surprisingly, some of the detected crops were not grown in the Kushiro and Nemuro regions. There was no significant difference in crop intake status in winter and that in other seasons for most of the crops. Corn (28.0%), soybeans (8.0%), wheat and beet (4.0%) were detected in crane chicks in summer, though the detection rates were generally lower than those in flying cranes. Alfalfa, which is not grown in eastern Hokkaido but is used in some cattle feed, was detected in some cranes. Rice, buckwheat, adzuki beans, common beans, potatoes and carrots were not detected at any life stage, indicating the preferences of red-crowned cranes. The results suggest that red-crowned cranes in Hokkaido are dependent on dairy farmers for their feed supply.

## 1. Introduction

The red-crowned crane (*Grus japonensis)* is protected in some countries of Far East Eurasia, and is designated as a Special Natural Monument and endangered species in Japan [1]. It is estimated that about 1800 red-crowned cranes inhabit the southeastern area of Hokkaido [2]. Red-crowned cranes have four developmental stages: chick (lacking the ability to fly), juvenile (with their parents within one year after birth), subadult (away from their parents without fertility) and adult (with fertility) [3,4].

There are two major populations of red-crowned cranes in the world: a mainland population in the Asian Far East and an island population in Hokkaido, Japan. Red-crowned cranes in the island population are residents and raise their offspring in summer, and spend winter around five major feeding stations in southeastern Hokkaido (four in the Kushiro area and one in the Tokachi area) in which corn is supplied by humans. The supply of corn is funded by the Ministry of the Environment, Japan (MOEJ) [4]. Some farmers also feed cranes with corn in minor private stations in winter [4]. A decision made by the MOEJ to gradually reduce the supply of dent corn in order to disperse the cranes may result in a change in their feeding status and influence their survival rate [5].

Observations using binoculars revealed that red-crowned cranes are omnivores that feed on fish, insects, frogs, spiders and various types of grain [6,7]. As for prey, the observations were roughly confirmed by high-throughput sequencing (HTS)-based metabarcoding with DNA extracts of scatological samples [8]. As for crops, it has been reported that red-crowned cranes feed on buckwheat, Chinese cabbage, cabbage, radish, and soybeans, in addition to corn in Hokkaido, although it is not clear whether they were observed eating in the field or were fed in captivity [6]. As for plants other than crops, cranes feed on grasses and clover in pastures and other fields, such as timothy, poaceae weeds, riparian weeds, nuts and the fruit of trees in Hokkaido [6,9]. In Korea and also in Hokkaido, artificial wetlands including rice paddies are used by red-crowned cranes both as night roosts and as foraging areas [10,11,12]. In China, damage to crops caused by red-crowned cranes was estimated to be 2% to 3% of the cultivated area of wintering areas [13].

The main habitats of red-crowned cranes in the island population are three areas in southeastern Hokkaido (Kushiro, Nemuro, and Tokachi). Kushiro and Nemuro are mainly dairy farming areas and the main crop is grass and some dent corn, while Tokachi is one of the representative areas for the cultivation of field crops. Various crops including soybeans (*Glycine max*), common beans (*Phaseolus vulgaris*), potatoes (*Solanum tuberosum*), sweet beet (*Beta vulgaris subsp. vulgaris*), and wheat (*Triticum* spp.) are grown on a large scale in Tokachi [14]. Red-crowned cranes forage on freshly planted corn, and feeding damage caused by cranes is now also officially recognized [15]. The Hokkaido government provides annual reports on agricultural damage caused by wild animals, and, on the basis of a survey of farmers, damage to potatoes, dent corn, sweet corn, and wheat caused by red-crowned cranes has also been recorded. This may not a large portion of the total damage in Hokkaido (about $40,000/year; only 0.11% of total damage caused by wildlife); however, the psychological damage to farmers should be considered [16]. Furthermore, as the habitat occupied by red-crowned cranes is expanding to central Hokkaido due to overcrowding of cranes in southeastern Hokkaido [5,17], there is also a concern about possible feeding damage to crops other than those in southeastern Hokkaido.

Most of the red-crowned cranes in Hokkaido now live in or near human settlements. For cranes and people to coexist, the problem of crane feeding-damage must be addressed [4,17]. HTS-based metabarcoding should be useful for determination of the crops cranes as well as other bird species feed on [18,19]. However, Kataoka et al. [8] reported that universal primers targeting animals in general may not be effective in detecting some animal species, for unknown reasons. In this study, we analyzed DNA extracts of scatological samples from flying cranes and chicks based on HTS of amplicon products using crop-specific primers. We also estimated the feeding preferences of red-crowned cranes and what crops they may affect in the future. The aim of this study was to obtain quantitative data on the crops actually consumed by wild red-crowned cranes in Hokkaido as a basis for developing effective countermeasures.

## 2. Materials and Methods

### 2.1. Samples and DNA Extraction

Contents of the small intestines of 60 flying cranes (adult, subadult and juvenile cranes) were obtained (Appendix A). Plastic spoons, gloves and surgical knives that were used to obtain intestinal contents were changed for each collection. Cranes were found dead in 2006–2021 in the Kushiro area (Kushiro City Hall 42.985, 144.381), Nemuro area (Nemuro City Hall 43.330, 145.583), Tokachi area (Obihiro City Hall 42.924, 143.196) and Abashiri area (Abashiri City Hall 44.021, 144.274) (Figure 1A), and they had been kept in a freezer in Kushiro Zoo, Kushiro, Hokkaido. An adult crane found dead in Erimo was also included. The habitat of the red-crowned cranes examined in this study belongs to the subarctic zone, with an average temperature of 6.45 °C (minimum: −10.5 °C, highest: 22.7 °C) and an average precipitation of 934.8 mm from 1981 to 2010 [20] (Temperature 2023). The vegetation was described in detail by Masatomi and Masatomi [4]. At the five major feeding stations, dent corn is fed from December to March. That period was therefore designated as the feeding period and the other months were designated as the non-feeding period. Intestinal contents were collected from the crane carcasses. With permission from the Japanese Ministry of the Environment (MOEJ: Tokyo, Japan) (1704261, 1704281, 1806126, 1806141, 1806151, 1906191), feces were collected from 25 chicks in June and July of 2016–2018 during banding (Appendix A, Figure 1B). Intestinal contents were collected sequentially with disposable plastic spoons. Feces were collected over a large clean plastic sheet to avoid contamination of weeds on the ground. Plastic sheets were washed with a large amount of water and dried indoors after each use to avoid possible contamination of the last feces sample. Samples were kept in plastic tubes for a few hours until freezing at −20 °C.

In order to confirm the presence of residual corn and other crops, feces from dairy cows (Holsteins) raised in Rakuno Gakuen University were collected using disposable plastic spoons and the contents were observed under a stereomicroscope.

Genomic DNAs were extracted from about 200 mg of scatological samples (feces and intestinal contents) (wet mass) using a DNeasy Plant Mini Kit (Qiagen, Venlo, Netherlands) according to the manufacturer’s instructions, based on the use of silica gel membranes that bind DNA with a high salt concentration buffer and elute DNA with a low salt concentration buffer. Extracted DNA was diluted in TE buffer (10 mM Tris-HCl and 1 mM EDTA, pH 7.5) and stored at −20 °C until use.

### 2.2. High-Throughput Amplicon Sequencing

The trnL region of an intergenic sequence of the chloroplast genome was used as a genetic marker. A nested PCR protocol was used to proliferate a very small amount of intact genomic DNA for PCR in scatological samples. The first-round PCR was carried out with a DNA sample prepared as described in the previous section using Tks Gflex DNA Polymerase (Takara Bio, Kusatsu, Japan). We used stepdown PCR consisting of a stepdown procedure (94 °C for 60 s as initial denaturation, 3 cycles of denaturation at 98 °C for 10 s, annealing at 65 °C for 15 s, extension at 68 °C for 30 s; 3 cycles of denaturation at 98 °C for 10 s, annealing at 59 °C for 15 s, and extension at 68 °C for 30 s; 30 cycles of denaturation at 98 °C for 10 s, annealing at 53 °C for 15 s, and extension at 68 °C for 30 s). The first PCR products were used for the nested PCR as templates utilizing the same stepdown protocol. Primers used for the first PCR and nested PCR are listed in Appendix A. Sixteen crops were selected as targets according to a feeding crop list obtained from observations with binoculars of wild and captive red-crowned cranes and the planting situation in eastern Hokkaido [6,14].

For the addition of Illumina adapters and barcodes to the nested PCR products to identify samples from individual cranes, according to the 2-Step PCR Amplicon Library Preparation (Preparing Dual Index Amplicons Library for the Illumina MiSeq System) (Illumina, San Diego, CA, USA), additional PCR with the following primer sets was carried out: 5′-End of forward primers (Amplicon F primers) with the addition of “TCGTCGGCAGCGTCAGATGTGTATAAGAGACAG” and 5′-end of reverse primers (Amplicon R primers) with addition of “GTCTCGTGGGCTCGGAGATGTGTATAAGAGACAG” were used (Appendix A). Additional PCR (8 cycles) was carried out with primers using the Nextera index kit (Illumina) to add identification sequences for the indexing of individual crane samples. These PCR reactions were performed in the same conditions as those described above. All of the PCR products were electrophoresed in 1% agarose gel, and ethidium bromide-stained bands were observed using a gel documentation system (WSE-5400: ATTO, Tokyo, Japan).

The remaining PCR products (amplicons) were then purified with a Fast Gene TM Gel/PCR Extraction kit (NIPPON Genetics, Tokyo, Japan), and amplicons were mixed for each scatological sample before high-throughput amplicon sequencing (HTAS). After purification with AMpure XP (Beckman Coulter Life Sciences, Brea, CA, USA) and validation and quantification of the prepared samples with TapeStation (Agilent Technologies, Santa Clara, CA, USA), mixed amplicons were sequenced with Illumina MiSeq PE (100 thousand reads/sample). All of these procedures were carried out according to the manufacturers’ instructions unless otherwise noted.

Raw reads were trimmed with two programs, Cutadapt (https://cutadapt.readthedocs.io/en/stable/) and Trimmomatic (https://github.com/usadellab/Trimmomatic). Since sequencing was carried out with forward (Read1) and reverse (Read2) primers, respectively, the same sequences contained in the data were counted for the read pairs that were linked by Read1 and Read2 with fastq-join (https://expressionanalysis.github.io/ea-utils/) to calculate read counts of each sample. A read count with less than 0.1% of total read counts or 10 was discarded to eliminate false positives resulting from index hopping [21]. Linked-read sequences were confirmed by a BLAST search in GenBank (https://blast.ncbi.nlm.nih.gov/Blast.cgi). All of these websites used for statistical analysis were accessed on 2 September 2023.

### 2.3. Specific PCR

In order to detect alfalfa, conventional nested PCR reactions were carried out with primer sets specific for alfalfa in this study (Appendix A). Since we did not find clear positive bands in agarose electrophoresis by the first PCR, first PCR and nested PCR procedures with Tks Gflex DNA polymerase (Takara Bio, Kusatsu, Japan) were carried out as described in Section 2.2. Some bands were extracted from the agarose gel and used for conventional Sanger sequencing to confirm the specificity with an ABI PRISM 310 Genetic Analyzer (Thermo Fisher, Waltham, MA, USA) after reactions with a Big Dye Terminator v3.1 Cycle Sequencing Kit (Thermo Fisher, Waltham, MA, USA).

### 2.4. Statistical Methods

Crop intake status between two groups was compared by Pearson’s chi-squared test. The numbers of crop species consumed by the cranes were compared by Poisson regression and the likelihood ratio test (R version 4.2.2, R Core Team 2022) [22].

## 3. Results

### 3.1. General Aspects of the Feeding Status of Flying Cranes

Some of the eight crops examined (except lettuce/prickly lettuce, see below) were detected in most of the flying cranes (Figure 2, Appendix A). The percentage of red-crowned cranes for which some crops were detected was 75% (45 out of the 60 cranes) and the average number of crop species detected in the crane was 0.95 (0–3 crops) throughout the year (Appendix A). Since the read sequences for cabbage and broccoli (*Brassica oleracea var. italica*) were perfectly matched, they are indicated as cabbage/broccoli (in Figure 2 and all other figures and tables). The read sequences of lettuce and prickly lettuce (*Lactuca scariola* L.) were also the same.

The dominant crop detected was corn throughout the year (61.7%, 37 out of 60). The rank order of detection rates for other crops was soybeans (8.3%, 5 out of 60) = wheat > tomatoes (6.7%, 4 out of 60) > Chinese cabbage (5.0%, 3 out of 60) = barley > radish (1.7%, 1 out of 60) = cabbage (or broccoli) (Appendix A). Eight crane samples contained lettuce or prickly lettuce.

Rice (*Oryza sativa subsp. japonica*), buckwheat (*Fagopyrum esculentum*), adzuki beans (*Vigna angularis*), common beans (*Phaseolus vulgaris* L.), potatoes (*Solanum tuberosum*), carrots (*Daucus carota subsp. sativus*) and (sugar) beet (*Beta vulgaris*) were never detected in flying cranes (Figure 2), although these crops were grown in the crane habitat, especially in the Tokachi and Okhotsk regions (Appendix A). Since the read sequences of lettuce and prickly lettuce, a weed also found in eastern Hokkaido [23], completely matched, these were not considered a crop.

### 3.2. Comparison of the Feeding Status of Flying Cranes in the Feeding Period and in the Non-Feeding Period

Since dent corn is supplied for cranes in winter, detection rates of crops were summarized separately for April to November (non-feeding period) (Figure 2A) and December to March (feeding period) (Figure 2B). Detection rates of crops in the non-feeding and feeding periods were 73.0% (27/37) and 83.6% (19/23), respectively. The average numbers of crops detected were 0.95 (0–3 crops) in the non-feeding period and 1.00 in the feeding period (Appendix A). There was no significant difference in the distribution of ingested crops between the feeding period and non-feeding period according to the results of the Poisson regression and likelihood ratio test (X^2^ = 0.047, df = 1, *p* = 0.829). However, detection rates differed more than twofold between the feeding and non-feeding periods. For example, the detection rate of tomatoes in the non-feeding period was more than twice that in the feeding period. Conversely, Chinese cabbage and soybeans tended to be more abundant in the feeding period. Lettuce or prickly lettuce also tended to be more abundant in summer.

The rank order of detection rates in the non-feeding period (N = 37) was corn (62.2%, 23 out of 37) > wheat (8.1%, 3 out of 37) = tomatoes > barley (5.4%, 2 out of 37) = soybean > radish (2.7%, 1 out of 37) = cabbage (or broccoli) = Chinese cabbage. Lettuce/prickly lettuce was more frequently detected than some vegetables (16.2%, 6 out of 37). The order of detection rates in the feeding period (N = 23) was corn (60.9%, 14 out of 23) > soybean (13.0%, 3 out of 23) > Chinese cabbage (8.7%, 2 out of 23) > tomatoes (4.3%, 1 out of 23) = barley = wheat. Radish and cabbage were not detected in winter. The detection rate for lettuce/prickly lettuce was 8.7%, which was slightly lower than in the non-feeding period. Despite feeding corn, the detection rate of corn in the feeding period was almost the same as that in the non-feeding period. This is clearly demonstrated by the monthly ingestion status of corn shown in Appendix A.

### 3.3. Comparison of the Feeding Status of Flying Cranes in Kushiro and Nemuro Regions and That of Flying Cranes in Tokachi and Okhotsk Regions

Detection rates of crops in the Kushiro and Nemuro regions (known for dairy farming) and the Tokachi and Okhotsk regions (known for upland farming) in the non-feeding period were 81.8% (18/22) and 57.1% (8/14), respectively (Table 1 and Table 2).

The rank order of detection rates in the Kushiro and Nemuro regions in the non-feeding period (N = 22) was corn (72.7%, 16 out of 22) > wheat (9.1%, 2 out of 22) = soybean = tomatoes > barley (4.5%, 1 out of 22) (Table 1). In the Tokachi/Abashiri regions, on the other hand, corn was also the most frequently detected crop during the feeding period, but the detection rate was much lower than that in the Kushiro/Nemuro regions (42.9%, 6 out of 14). Many other crops were also detected in the Tokachi/Abashiri regions. Barley, wheat, radish, cabbage/broccoli and Chinese cabbage were each detected once (7.1%) (Table 2). There was no significant difference in the distribution of ingested crops between the Kushiro/Nemuro regions and Tokachi/Abashiri regions according to the results of the Poisson regression and likelihood ratio test (X^2^ = 10.505, df = 9, *p* = 0.231).

Lettuce or prickly lettuce was detected in the Kushiro/Nemuro regions (22.7%, 5) but was detected at a lower rate in the Tokachi/Abashiri regions (14.3%, 2) in the non-feeding period.

### 3.4. Intake of Grains and Vegetables by Chicks

The feces of crane chicks collected in June and July were examined. The detection rate of crops was 40.0% (10/25) and the average number of crop species detected was 0.40 (0–3 crops) (Appendix A). The average number of crop species detected was significantly smaller than that in flying cranes (chicks vs. feeding period, X^2^ = 5.388, df = 1, *p* = 0.020; chicks vs. non-feeding period, X^2^ = 5.349, df = 1, *p* = 0.021: according to the likelihood ratio test). Both values were lower than those in flying cranes. The rank order of detection rates (N = 25) was corn (28.0%, 7) > soybeans (8.0%, 2) > wheat (4.0%, 1) = beet (Figure 3, Table 3). Corn was only detected in July samples (Table 3). Three chicks showed lettuce or prickly lettuce. In addition to the crops that were not detected in flying cranes (rice, buckwheat adzuki beans, common beans, potatoes, carrots), barley, radish cabbage/broccoli, Chinese cabbage and tomatoes were not detected in chicks. However, a chick in Nemuro area showed beet in its feces.

Detection rates of crops in the Kushiro and Nemuro regions (well-known for dairy farming) and the Tokachi area were 41.2% (7/17) and 37.5% (3/8), respectively (Table 3). Both detection rates were similarly low when corn was excluded (Kushiro and Nemuro: 17.6%, Tokachi: 12.5%). Soybeans and wheat were both detected in the Kushiro area and soybeans were detected once in the Tokachi area. Sugar beet (beet) was detected in a chick in Nemuro, the only crane including flying cranes.

Lettuce/prickly lettuce (N = 3) was detected only in the Kushiro area, although there are extensive lettuce farms in some parts of the Tokachi area (such as Makubetsu City) (Table 3).

Figure 1B No. 18 (Banding No. 311) and No. 22 (326) show beet and wheat (Table 3); however, these crops were not grown within a 5 km radius of where these chicks were captured. Similarly, Figure 1B No. 21 (324) and No. 23 (334) shows lettuce or prickly lettuce, although lettuce was also not grown within a 5 km radius of where these chicks were captured.

### 3.5. Detection of Corn in Daily Cattle Feces and Alfalfa in Intestinal Contents of Red-Crowned Cranes

Corn, barley, wheat and soybeans are major components of the concentrate mixture in Hokkaido [23]. We confirmed corn grains in cow feces (Appendix A).

We focused on alfalfa, which is not grown in the red-crowned crane habitat in Hokkaido, but is contained in hay purchased from overseas and also in concentrate mixture as hay cubes. As shown in Appendix A, alfalfa was detected in six cranes by conventional PCR with specific primers. The amplicon was that of alfalfa, confirmed by Sanger sequencing.

## 4. Discussion

### 4.1. General Features

Although the amount of data is not large, these data provide the first quantitative data on crop consumption by red-crowned cranes in eastern Hokkaido that were obtained by amplicon analysis with HTS of genomic DNA extracted from scatological samples and PCR products of representative crops. Although the results of field-observation feeding surveys [6,7] have been reported, it is difficult to distinguish whether the cranes are feeding on plants or feeding on insects or other organisms that are lodged in the plants. Previously reported information on the ingestion of crops also included cases in which crops were given to captive individuals [6]. There is a concern that the contents of the digestive tracts of dead birds and healthy birds are different, and that might have affected the results obtained. However, it is unlikely because the effects of the conditions of scatological samples may be limited to the percentage of the crane population for which the crop was detected.

The results of this study revealed that red-crowned cranes feed on various kinds of crops. The crop intake rate was 76.7% in all of the 60 flying cranes throughout the year. Even in the non-feeding period (April–November), the crop intake rate was 73.0%. In the non-feeding period, an average of 0.95 crop species were eaten by each individual crane, with a maximum of three species detected. No relevant overall seasonal or regional differences in intake content and frequency were observed.

Of the crops examined, rice, buckwheat, adzuki beans, common beans, potatoes and carrots were never detected in either flying cranes or chick cranes. Kobayashi et al. [6] carried out observations and interviews and reported that wild red-crowned cranes ingested potatoes and carrots. Both crops are grown in Tokachi and Abashiri, areas with potato production in Japan [14]. In Amount of Damage by Birds and Animals in Hokkaido, 2022 [15], which was based on farmers’ reports, damage to potatoes was also reported. Although red-crowned cranes did not feed on potatoes, the commercial value of potatoes was reduced by cranes digging up the soil in the fields. Tokachi is one of the largest production regions for adzuki beans in Japan, but there do not seem to be any records of consumption of adzuki beans by red-crowned cranes, and consumption of adzuki beans was also not found in our study. DNA sequences of our reads of lettuce are exactly the same as those of wild prickly lettuce. However, these were detected at a high rate in flying cranes (15%, 9/60). Lettuce/prickly lettuce was frequently detected in the Kushiro region, where lettuce is not grown, and was detected more frequently in summer than in winter, suggesting that prickly lettuce might have been detected in most cases.

### 4.2. Juvenile, Subadult and Adult Red-Crowned Cranes

It was confirmed that corn is the most commonly consumed food for all 15 species of cranes, followed by wheat and kaoliang (*Sorghum bicolor* (Linnaeus) Moench) [24]. Corn, including sweet corn for humans and dent corn for dairy cows, is grown more or less throughout the crane habitat. We were unable to distinguish between these two types of corn in this study since no methods have been reported for damaged DNA, such as DNA extracts from scatological samples. Corn is sown in late May or early June and harvested in August or September in southeastern Hokkaido. Although it was quite natural that corn would be detected in the samples obtained in the feeding period from December to March, corn detection rates were not different in the non-feeding period (62.2%, 23/37) and the feeding period (60.9%, 14/23). Thus, 39.1% (9/23) of the individuals had no corn detected in the feeding period. This is in contrast to the speculation that red-crowned cranes in Hokkaido depend solely on human-supplied corn in winter. Thus, it is thought that some cranes may be dependent on other feed sources, at least on some days. This issue needs further study, including interviews with farmers and a field survey on the possible expansion of winter foraging areas due to recent global warming.

The main component of the total mixed ration for dairy cattle is corn, to which barley, wheat (bran), and soybeans (soybean cake) are also added [25]. Red-crowned cranes appear in barns and peck at compound feed for dairy cows [15]. If dairy cows ate compound feed, corn, wheat, and barley or at least some of these should have been detected at the same time, but such cases were rarely observed in this study. Furthermore, detection rates of barley (4.5% (1/22) in the Kushiro–Nemuro region vs. 7.1% (1/14) in the Tokachi–Okhotsk region), wheat (9.1% (2/22) vs. 7.1% (1/14)) and soybeans (9.1% (2/22) vs. 0% (0/14)) were comparable in these two regions. These crops are never or rarely grown in the Kushiro–Nemuro region, suggesting that these grains were not derived from farm fields. Alfalfa, which is not grown in Hokkaido but is a component of imported hay and concentrate mixture, was detected in five flying cranes using conventional nested PCR. These observations suggest that cow feces compost in the barn could be the major feeding area. The compost contained grains (mainly corn) that were not digested by the dairy cows.

The cultivation region of tomatoes is almost limited to Shiranuka in the Kushiro area in summer (Appendix A). However, tomatoes were detected in samples from Akkeshi Town and Tsurui Village in the Kushiro area. Tomatoes were found in a sample from Taiki Town, the Tokachi area in winter (early March). Two of the three cranes in which Chinese cabbage was detected were collected in the Kushiro area in winter. Thus, it is unlikely that these detected vegetables were from farmlands. Since it has been reported that red-crowned cranes can eat buckwheat [6], which was not detected at all in this study, it is not clear whether they were observed eating in the field or were fed in captivity [6].

Taken together, the results indicate that most red-crowned cranes ingest mainly corn and ingest other grains and vegetables to some extent. It is unlikely that the origins of most crops were farmlands, since the feeding rates of these crops did not differ between the feeding and non-feeding periods, or between the dry-field region (Tokachi and Abashiri) and dairy region (Kushiro and Nemuro). Naturally, private vegetable gardens are one of the potential feeding sources, but there must be some other feeding sources, since many vegetables were detected from late fall to early spring. As far as we know, no damage to vegetable gardens has been reported. Most dairy farmers dispose of kitchen scraps in compost piles or other areas around the barn [26]. Thus, human leftovers discarded by farmers may be an important feeding source for cranes. However, since rice, which should be present in farmers’ leftovers, was not detected at all, it is likely that cranes do not like rice. We cannot conclude at present whether red-crowned cranes eat rice or not because red-crowned cranes in Korea and Hooded cranes and White-naped cranes in Kyushu, Japan, feed on sprouts of rice [27] and other poaceae plants (Job’s Tears *Coix lacryma-jobi*) [28]. Additionally, it was reported that cranes showed high adaptability to foraging on different kinds of agricultural crops depending on their availability in the landscape [24]. Since rice is the largest crop in all areas of Japan except southeastern Hokkaido, we should take this propensity into account.

The digestive system of cranes is not suited to digesting high-fiber plant-foods, and cranes are dependent on prey such as insects and fish and plants with low fiber. Therefore, cranes, including red-crowned cranes, prefer animal foods as well as low-fiber, high-protein plants such as plant seeds [9,29,30]. An exception is soybean, which, despite its high crude protein content, is not preferred because biochemicals in raw soybeans inhibit nutrient-assimilation in the digestive system of cranes [31]. This may explain why soybeans were not detected so often in red-crowned cranes in Hokkaido, even though soybeans are a major crop in Tokachi [14]. Although further study is needed, this may also be related to the fact that other legumes, such as adzuki, were not detected.

Like other cranes, red-crowned cranes have adapted to changes in the environment [28]. For example, the expansion of agricultural land in China has led to the shrinkage of wetlands in red-crowned cranes’ breeding and stopover areas [24]. Similarly, in Hokkaido, the area of wetlands continues to shrink due to the development of agriculture and other industries [17], making red-crowed cranes dependent on dairy farmers and corn supplied by humans [4]. Naturally, changes in the crops commonly used in an area could drastically alter the feed available to cranes. In the case of the sandhill crane *Grus canadensis*, the shift from corn to soybean cultivation has resulted in cranes flying longer distances to feed, and has reduced their body fat reserves [31,32]. However, Kushiro and Nemuro, the major red-crowned crane habitats in Hokkaido, are dominated by grasslands and some dent corn fields and Tokachi and Okhotsk are field-crop areas; but our study showed that cranes in Hokkaido are largely dependent on corn derived from compost piles in both Kushiro/Nemuro and Tokachi/Okhotsk, unlike cranes in other regions, which may not be significantly affected by changes in crop content. Currently, red-crowned cranes in Hokkaido do not need to store energy since they do not migrate over long distances [4].

### 4.3. Chick Cranes

Chicks’ feces were collected in only June and July, and it is therefore possible that all of the crops were field-derived, unlike those in the intestinal contents of flying cranes. In addition, since chicks are flightless and only walk, they can only travel within a very limited area, making it easier to identify the source of their intake compared to that in flying cranes.

Corn was also the most frequently detected crop in the feces of chicks. Considering the season of sampling, it is possible that the corn detected in the feces was derived from corn grown in the field. In addition, all seven samples of feces from chicks in which corn was detected were collected in July. Although red-crowned cranes are often seen feeding on newly sprouted corn [6], corn is planted in early June in southeastern Hokkaido, which rules out the possibility of feeding on corn sprouts in the field. Since samples of corn and other grains including barley, wheat and soybeans were never found in chicks at the same time, there was no evidence of chicks feeding on compound cattle feed in this study. Alfalfa was also detected in the feces of three chicks, suggesting that some chicks also came to compost heaps in dairy farms.

Sugar beet was also detected in Bekkai Town in the Nemuro area, where sugar beet is not grown but is included in compound cattle feed as beet pulp, which may have been ingested [33]. There must be a source of ingestion other than the field since two of the eleven samples of chick droppings with a confirmed crop (beet and wheat) had no corresponding crops within a 5 km square area around the sampling site. Taking these facts into consideration, it is thought that chicks also use compost and other sources around the barn as important feed sources. The detection rate of crops tended to be lower in chicks than in flying cranes in this study, possibly due to a limited range of traveling for chicks compared to that for flying cranes. In combination with the fact that there is no evidence for ingestion of compound cattle feed by chick cranes, it is also speculated that chicks may be reluctant to enter the barn where they cannot escape immediately when they encounter humans, since they cannot fly.

According to Kataoka et al. [8], who examined the prey of red-crowned cranes in Hokkaido, the detection rates of insects and fish, including other feed items, in chicks were clearly lower than those in flying cranes. Given that red-crowned cranes are highly dependent on barns and their surroundings, including compost heaps, it is possible that chicks forage less than do flying cranes.

### 4.4. Extent of Crop Damage Caused by Red-Crowned Cranes

It is thus thought that red-crowned cranes, including chicks, use barns as important feeding sites but rarely ingest compound feed for dairy cows. Therefore, red-crowned cranes might not cause significant damage to crops or a significant economic loss for farmers in southeastern Hokkaido, at least in terms of feeding damage. A bigger problem might be stress for cows when cranes enter the barn, which might reduce milk production [17]. Crops that red-crowned cranes appear to dislike, including adzuki beans, carrots, barley, buckwheat, and other crops, were not a problem for farmers. However, many of the samples used in this study for dietary analysis were collected in the Kushiro and Nemuro regions, where field crops are not grown. Therefore, further study is needed to determine the feeding preferences of cranes using more samples from the Tokachi and Abashiri regions, where field crops are more prevalent. Additionally, consideration should be given to the possibility that cranes cause damage to fields and crops by kicking around the crops as in the case of potatoes [34]. It is indeed a problem that cranes feed on newly sprouted corn in spring [6]. Although 9,10-anthraquinone (AQ) is commonly used as a low-toxicity taste inhibitor for other wild birds worldwide [35], its cost-effectiveness must also be considered. The damage to farms from grazing on fresh shoots during the corn-planting season in June and the destruction of crops and farms without ingestion may be serious problems. Setting up nets on farms may currently be the most practical response [34]. Red-crowned cranes already breed in the central area of Hokkaido, which is also an agricultural area (in particular, for rice cultivation [36]), and this could cause crop damage on a large scale.

## 5. Conclusions

Crop damage caused by red-crowned cranes has been frequently observed in the Tokachi and Abashiri regions [17,34]. However, the results of this study suggest that most of the crops detected in the scatological samples of red-crowned cranes were derived from undigested corn or other grains for dairy cattle and farmers’ leftovers, and that there were relatively few cases of consumption of crops in farmlands. The concentrate mixture may not be a major source of feeding for cranes, since the main ingredients (corn, barley, wheat, soybeans) were rarely detected in the same crane. The results of this study provide basic data for taking effective measures against possible future problems. It is important to interview farmers and private households feeding cranes, since there are many private feeding stations in southeastern Hokkaido. It should also be considered that recent global warming may have increased the number of places where cranes can find sufficient food without having to rely on feeding stations in winter [17].

## Figures and Tables

**Figure 1 animals-13-03167-f001:**
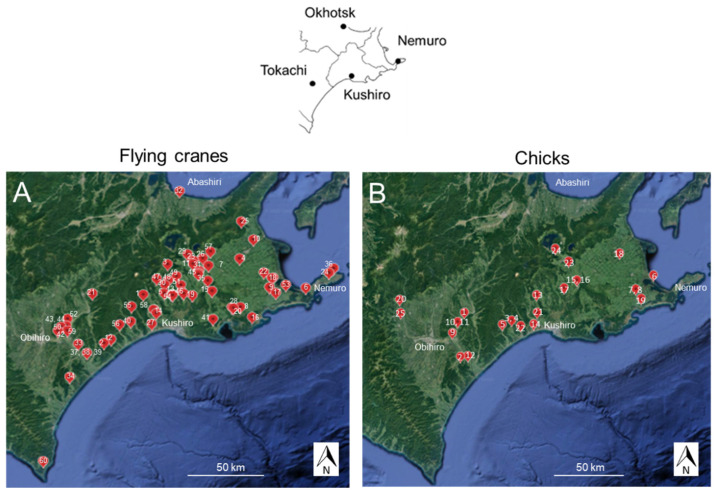
Sampling sites of bodies of flying red-crowned cranes (**A**) and chicks (**B**) in southeastern Hokkaido, Japan. (**A**) Bodies of adult and subadult cranes found dead in the field (N = 60). (**B**) Feces of red-crowned crane chicks (N = 25). Collection sites of bodies were plotted on a map that was made using Google Earth. The numbers in figures indicate sample numbers in Appendix A. The upper right inset shows the Hokkaido Branch area in southeastern Hokkaido. The location of the city hall of the central city of each branch is indicated by a black dot (Obihiro for Tokachi branch, Kushiro for Kushiro branch, Nemuro for Nemuro branch, Abashiri for Okhotsk branch).

**Figure 2 animals-13-03167-f002:**
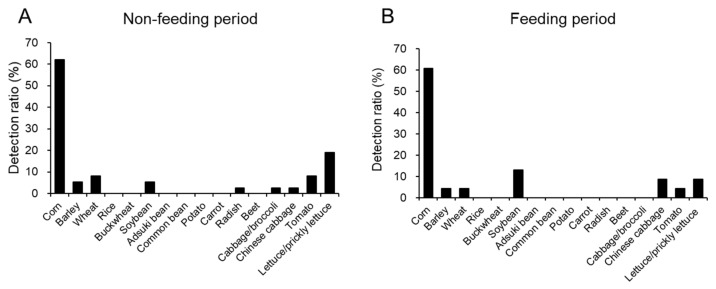
Detection rates of sixteen crops in flying red-crowned cranes in the feeding and non-feeding periods. Intestinal contents of flying red-crowned cranes (juveniles, subadults and adults) were used for analysis. Data for Non-feeding period (**A**) (N = 37) and feeding period (**B**) (N = 23) are shown. One sample from an adult crane found dead in Erimo was included in the non-feeding period. Cabbage/broccoli and Lettuce/prickly lettuce mean that read sequences of cabbage and broccoli and those of lettuce and prickly lettuce were the same.

**Figure 3 animals-13-03167-f003:**
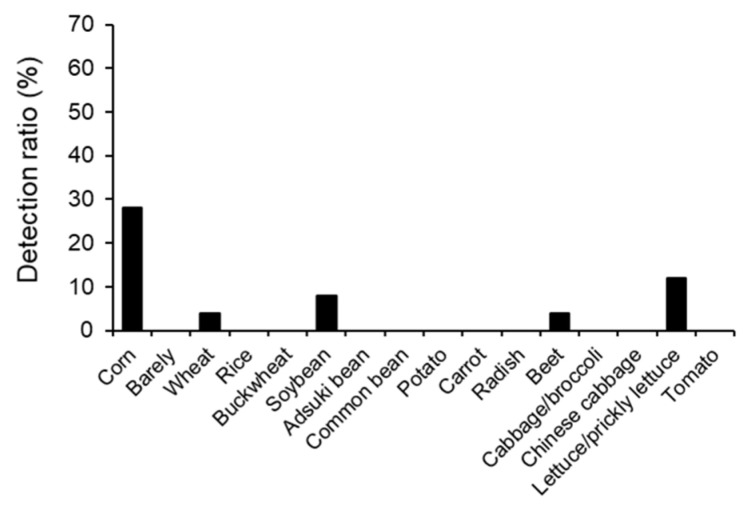
Detection rates of sixteen crops in red-crowned crane chicks in summer. Chick feces collected in June and July (N = 25) were used for analysis.

**Table 1 animals-13-03167-t001:** Assessment of crop consumption of individual red-crowned cranes during the non-feeding period: intestinal content analysis in Kushiro and Nemuro regions.

Figure 1A No.	Reference No.	Collection Date	Collection Site	Corn	Barley	Wheat	Soybean	Radish	Cabbage /Broccoli	Chinese Cabbage	Tomato	Lettuce /Prickly Lettuce
3	R143	28-Jun-2006	Shibecha	〇								
4	R146	19-Oct-2006	Bekkai	〇	〇							
5	R147	22-Oct-2006	Tsurui	〇								
6	R148	23-Oct-2006	Nemuro	〇								
9	R160	4-Apr-2007	Hamanaka	〇								〇
10	R161	9-May-2007	Nakashibetsu	〇								
11	R165	10-Jul-2007	Shibecha	〇		〇						
13	R176	20-Nov-2007	Tsurui				〇					
14	R178	27-Nov-2007	Kushiro									〇
18	R196	21-Jul-2008	Bekkai	〇								
22	R226	2-Jun-2009	Bekkai	〇								
23	R241	22-Nov-2009	Teshikaga	〇								
27	R251	24-May-2010	Shiranuka	〇			〇					
28	R262	29-Nov-2010	Akkeshi								〇	
40	R542	30-Apr-2020	Shiranuka	〇								
41	R547	29-Jul-2020	Akkeshi									〇
45	R551	4-Oct-2020	Shibecha	〇		〇						
47	R553	9-Oct-2020	Tsurui	〇								〇
48	R555	18-Oct-2020	Tsurui	〇							〇	
49	R556	26-Oct-2020	Tsurui	〇								〇

Individual status for crop detection in intestinal contents of flying red-crowned cranes found in Kushiro and Nemuro areas during the non-feeding period. Circle mark indicates that the crop was detected. Data are shown for 18 flying red-crowned cranes (subadults and adults) that showed any crops in their intestinal contents. Data were compiled only for cranes that were found in the Kushiro area and Nemuro area from April to November (total of 22 flying cranes).

**Table 2 animals-13-03167-t002:** Assessment of crop consumption of individual red-crowned cranes during the non-feeding period: intestinal content analysis in Tokachi and Okhotsk regions.

Figure 1A No.	Reference No.	Collection Date	Collection Site	Corn	Barley	Wheat	Soybean	Radish	Cabbage /Broccoli	Chinese Cabbage	Tomato	Lettuce /Prickly Lettuce
12	R175	16-Nov-2007	Urahoro	〇								
21	R225	21-May-2009	Honbetsu	〇								
32	R272	7-May-2011	Abashiri	〇	〇				〇			
33	R275	25-Jun-2011	Toyokoro	〇								
34	R281	2-Nov-2011	Taiki								〇	〇
37	R474	3-Sep-2018	Urahoro	〇								
39	R476	3-Sep-2018	Urahoro			〇		〇		〇		〇
59	R580	5-Jun-2021	Ikeda	〇								

Individual status for crop detection in the intestinal contents of flying red-crowned cranes found in the Tokachi and Okhotsk areas during the non-feeding period. Circle mark indicates that the crop was detected. Data are shown for eight flying red-crowned cranes (juveniles, subadults and adults) that showed any crops in their intestinal contents. Cranes with no crops were not included. Data were compiled only for cranes that were found in the Tokachi area and Okhotsk area from April to November (total of 14 flying cranes).

**Table 3 animals-13-03167-t003:** Assessment of crop consumption of individual red-crowned crane chicks: fecal analysis.

Figure 1B No.	Banding No.	Sex	Collection Date	Collection Site	Reg. Promo. Bureau	Corn	Wheat	Soybean	Beet	Lettuce /Prickly Lettuce
13	305	Female	4-Jul-2017	Kushiro city	Kushiro	〇		〇		
14	306	Male	4-Jul-2017	Shiranuka	Kushiro	〇				
16	308	Male	5-Jul-2017	Shibecha	Kushiro	〇				
18	311	Male	7-Jul-2017	Bekkai	Nemuro				〇	
21	324	Female	13-Jul-2018	Kushiro	Kushiro					〇
22	326	Female	7-Jul-2018	Shiranuka	Kushiro		〇			〇
23	334	Female	14-Jul-2018	Teshikaga	Kushiro	〇				〇
24	336	Female	14-Jul-2018	Teshikaga	Kushiro	〇				
5	272	Male	2-Jul-2016	Onbetsu	Tokachi	〇				
11	297	Male	24-Jun-2017	Ikeda	Tokachi			〇		
20	317	Female	9-Jul-2017	Shintoku	Tokachi	〇				

Individual status for crop detection in intestinal contents of red-crowned crane chicks found in Kushiro area and Nemuro area. Circle mark indicates that the crop was detected. Data are shown for 11 red-crowned crane chicks that showed any crops in their feces. Data were compiled for crane chicks that were found in Kushiro area and Nemuro area (17 chicks in total) and in Tokachi area (total of 8 chicks).

## Data Availability

The datasets generated during and/or analyzed during the current study are available from the corresponding author on reasonable request.

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
