# Peer review of "Analysis of Crop Consumption Using Scatological Samples from the Red-Crowned Crane Grus japonensis in Eastern Hokkaido, Japan"

_animals, 2023, doi:10.3390/ani13203167_

Round 1

Reviewer 1 Report

Line 174-175: the number "46" , how did you get it? From the table S5, we know the detected quantity of corn, soybean, wheat, tomatoes, chinese cabbage, barly, radish, cabagge are 37, 5, 5, 4, 3, 3, 1 and 1. The total is far than 46. Same as the number "0.95", how get it?

Line 321-323: why kobayashi et al said red-crowned cranes eat carrots and your research showed not? who is ture?

Line 344: the rate of no corn individuals was 39.1%,  is still quite high, why? How do you process in the further study?

Line 433-441: the contents were not the research results and it is recommended to include it in the discussion.

Author Response

Reply to Reviewer 1

1) Line 174-175: the number "46" , how did you get it? From the table S5, we know the detected quantity of corn, soybean, wheat, tomatoes, chinese cabbage, barly, radish, cabagge are 37, 5, 5, 4, 3, 3, 1 and 1. The total is far than 46. Same as the number "0.95", how get it?

Response: We are very sorry. We meant that “The percentage of individual red-crowned cranes for which some crop was detected was 75% (45 out of 60)”. Our apologies, 45 was correct, not 46. Since number of cranes that ate crop was 45, we have corrected it. The value of 0.95 is the average number of crops species detected from one crane. We corrected the other parts as well (L29-212). We highlighted modified sentences in red in the revised manuscript.

2) Line 321-323: why kobayashi et al said red-crowned cranes eat carrots and your research showed not? who is ture?

Response: We did not know the reason actually. However, this may be related to the fact that they counted the case in which they eat carrots given to them at zoos, etc., other than their own observations.

3) Line 344: the rate of no corn individuals was 39.1%, is still quite high, why? How do you process in the further study?

Response: This is very surprising indeed. Identifying the causes of this problem is important but very difficult at this time. The following three points are worth mentioning.

1) We compared the present data with our previous study on the animal food intake of the same red-crowned cranes examined in this study (Kataoka et al., J Vet Med Sci. 84: 358-367. 2022.). Individual crane in which no corns were detected did not have a higher number of animal species detected than those in which corns were detected {Number of animal species detected: corn (-) 2.3±0.7 (n=12) vs corn (+) 8.6±0.7 (n=23)}. We would like to compare the amount of intake, but the quantifiability of HTS using fecal samples is unreliable. 

2) It is possible that they may be eating some other crop or plant species in addition to the ones examined in this study. It is possible to examine the number of plant species ingested by performing HST using multiple universal primers. Another problem is that the quantifiability of HTS using fecal samples is not reliable. However, due to snow and icing in winter, it is unlikely that there are many calorie-rich wild plants in southwestern Hokkaido during the winter season.

 3) Recent global warming may have expanded the number of feeding areas available for winter use. Interviews with farmers and field surveys are needed.  

 Some of these considerations (especially the third) were summarized in the revised manuscript (L408-410).

4) Line 433-441: the contents were not the research results and it is recommended to include it in the discussion.

Response: We moved most sentences into L524-530 in Discussion except the last sentence, since other Reviewer requested some perspectives of this study with a couple of additional sentences.

Reviewer 2 Report

1- Is the main purpose of this research to investigate damage to the crop by the cranes? Or is this endangered bird species also important?

2- If the damage to the crop is significant, the number of the total bird population and the amount of damage should be estimated.

3- Line 40: Food and feed are used synonymously, while for the purpose of differentiation, it is better to use the feed term for birds and food for humans.

4- No statistical analysis has been done to draw conclusions. For example, the correlation between the age (chick, juvenile, adult, ...) of the bird, the season, the geographical location, and the type of crop consumed, which indicates the consumption of crop by the birds due to its availability or the bird's preference.

5- Line 96: There is a possibility that the contents of the digestive tract of dead and healthy birds are different, therefore it can affect on the obtained results and conclusion. So it is better to use faeces samples for statistical analysis and conclusion.

English is not fully fluent scientifically. Because such articles will be studied by people with different knowledge, information and different expertise.

Author Response

1) Is the main purpose of this research to investigate damage to the crop by the cranes? Or is this endangered bird species also important?

Response: As understood by Reviewer, the main purpose of this research to investigate damage to the crop by the cranes. However, our study is also significant as an example of crop damage caused by endangered and highly protected species. We added a special natural monument in Abstract in red (L21-22). Since the red-crowned crane is a highly protected bird, farmers are more annoyed than the pest birds.

2) If the damage to the crop is significant, the number of the total bird population and the amount of damage should be estimated.

Response: It is estimated that about 1,800 red-crowned cranes inhabit the southeastern area of Hokkaido. Hokkaido Government has announced that the total damage to farmers caused by red-crowned cranes is about $40,000/year, based on farmers' declarations. Hokkaido Government has not provided details, but damage to crops on farms is part of the problem. Since we suggested that the main foraging grounds for red-crowned cranes are compost heaps in this study, the amount of damage may not be very large. We added estimated number of cranes in Hokkaido and amount of damage by cranes in red (L51-52, L92). We think that red-crowned cranes might not cause a serious damage to crops in southeastern Hokkaido, at least in terms of feeding damage (L469-474).

3) Line 40: Food and feed are used synonymously, while for the purpose of differentiation, it is better to use the feed term for birds and food for humans.

Response: Thank you for your comments. We have corrected it and the others as Reviewer suggested.

4) No statistical analysis has been done to draw conclusions. For example, the correlation between the age (chick, juvenile, adult, ...) of the bird, the season, the geographical location, and the type of crop consumed, which indicates the consumption of crop by the birds due to its availability or the bird's preference.

Response: We performed Pearson's chi-squared test to compare the overall crop intake status between two groups (season, geographical location, flying cranes vs chicks). Distribution of number of crop species consumed per individual crane was compared by Poisson regression and Likelihood ratio test. We added a chapter about statistical analysis in Materials and Methods (L209-213), and added some descriptions in Results section (L253-255, L283-285, L312-314).

5) Line 96: There is a possibility that the contents of the digestive tract of dead and healthy birds are different, therefore it can affect on the obtained results and conclusion. So it is better to use faeces samples for statistical analysis and conclusion.

Response: Intestinal contents were obtained by flying cranes, which were dead by accidents including traffic accidents and collision with utility poles for most cases. If the condition differs between feces samples, it is expected to be limited to quantitative effects. We did not consider quantitative data on how much crop was being eaten. However, since Reviewer’s suggestion is important to consider, we have added a note about the possible effect of feces and intestinal contents on the detection rate (L369-372).

Reviewer 3 Report

Yokokawa et al.: Analysis of crop consumption using scatological samples from the red-crowned crane Grus japonensis in eastern Hokkaido, Japan 

Overall, the text is well written. However, it would gain from some language polishing, and I would like to demonstrate this by showing examples of just two sentences:  

Line 23: The original sentence: “They feed on plants including grains in addition to insects and fish as omnivores.” I would write as follows: “As omnivores, they feed on plants, including grains, insects, and fish.” 

Lines 46-48: The original sentence: “The red-crowned crane Grus japonensis is highly protected in some countries in and around Far East Eurasia as a special natural monument in Japan and an endangered bird species [1].” I would suggest changing “special” with “unique” as follow: “The red-crowned crane Grus japonensis is highly protected in some countries in and around Far East Eurasia as a unique natural monument in Japan and an endangered bird species [1].” 

Line 89: “Thus” replace with “In this study” 

Line 103: To be precise, replace “weight” with “mass”. These two parameters are not the same.  

Lines 190-191: I would suggest you continue with adding "out of" [(8.3%, 5 out of 60) = wheat > 190 tomatoes (6.7%, 4)] throughout the text like this: "(6.7%, 4 out of 60)".  

Lines 190-191: I would suggest you continue with adding "out of" [(8.3%, 5 out of 60) = wheat > 190 tomatoes (6.7%, 4)] throughout the text like this: "(6.7%, 4 out of 60)". The same is in line 207 and the text below, where you have (N = 37) as the total number of samples, and line 224, where your sample size is 24.  

Line 245: November (l 14 flying cranes). What is "I"? 

Line 256: n=25; while you use N in other parts of the Results. Please, be consistent!  

Why do you use curly brackets (braces) in lines 354-355?  

Overall, the text is well written. However, it would gain from some language polishing

Author Response

Overall, the text is well written. However, it would gain from some language polishing, and I would like to demonstrate this by showing examples of just two sentences: 

1) Line 23: The original sentence: “They feed on plants including grains in addition to insects and fish as omnivores.” I would write as follows: “As omnivores, they feed on plants, including grains, insects, and fish.”

Response: Thank you for Reviewer's kind suggestion. We replaced the sentence as suggested, as highlighted in red (L24-25).

2) Lines 46-48: The original sentence: “The red-crowned crane Grus japonensis is highly protected in some countries in and around Far East Eurasia as a special natural monument in Japan and an endangered bird species [1].” I would suggest changing “special” with “unique” as follow: “The red-crowned crane Grus japonensis is highly protected in some countries in and around Far East Eurasia as a unique natural monument in Japan and an endangered bird species [1].”

Response: We are very afraid that “special natural monuments” are officially used by the Japanese government.

3) Line 89: “Thus” replace with “In this study”

Response: We modified the sentence as suggested (L100).

4) Line 103: To be precise, replace “weight” with “mass”. These two parameters are not the same. 

Response: Thank you very much. We replaced the word as suggested (L136).

5) Lines 190-191: I would suggest you continue with adding "out of" [(8.3%, 5 out of 60) = wheat > 190 tomatoes (6.7%, 4)] throughout the text like this: "(6.7%, 4 out of 60)". 

Response: Thank you very much. We replaced the sentence as suggested (L235-236).

6) Lines 190-191: I would suggest you continue with adding "out of" [(8.3%, 5 out of 60) = wheat > 190 tomatoes (6.7%, 4)] throughout the text like this: "(6.7%, 4 out of 60)". The same is in line 207 and the text below, where you have (N = 37) as the total number of samples, and line 224, where your sample size is 24. 

Response: We modified the sentence as suggested (L260-265, L277-278).

7) Line 245: November (l 14 flying cranes). What is "I"?

Response: We apologize. We removed it.

8) Line 256: n=25; while you use N in other parts of the Results. Please, be consistent! 

Response: We are sorry. We used "N" through the manuscripts.

9) Why do you use curly brackets (braces) in lines 354-355? 

Response: This is because it contained the regular brackets inside. We replaced the curly brackets with the regular brackets.

Reviewer 4 Report

Abstract

- Please add numbers when you present your results.

- L 34-35: Please rephrase “Additionally, there was little difference between the crop feeding status of cranes in winter and other seasons”. What does it mean “little difference”?

Introduction 

- The introduction is incomplete. More comprehensive and recent literature review is needed.

- Please add previous studies about the diet of red-crowned cranes 

- L 57: The reference [4] is inaccurate. Please check and correct.

- L 57: add a comment (and a reference) on the decision of the MOEJ to gradually reduce the supply of dent corn in order to disperse the cranes, which may change their feeding status and may influence their survival rate.

- L 60-66: This is unnecessary information for this manuscript.

- L 63: Abbreviations should be defined when they firstly appear. Such as “HTS-based metabarcoding”

- L 69: add a reference.

- L 83-86: add a reference.

- L 89-92: please formulate a purpose of your research article.

Materials and methods 

 - L 96-97: please add for each area the geographical coordinates.

- L 97: give some information related to the study area (climate, temperature, precipitation, vegetation).

- L 101: add details regarding the collection of feces and the content of the small intestine. Explain the division of samples taking into consideration the feeding period.

- L 102: please add some details for samples preparation.

- L 103: remove “by”

- L 103: add city and country for Qiagen.

- L 170: The section regarding data analysis is missing. There is no description of statistical analysis.

- Dairy cattle feces are mentioned in this study. Please add the details related to the collection of these samples and describe the purpose of cattle feces collection.

Results

- L 182: the sentence is repeated. Please remove it.

- L 183-187: the paragraph should be moved to Materials and methods

Discussion 

There is little in the way of insightful discussion in this section.

- L 319-320: “No relevant seasonal or regional differences in intake content and frequency were observed”. No differences were observed based on a statistical analysis?

- L 346: “Thus, 39.1% (9/23) of the individuals had no corn detected in the feeding period. This is in contrast to assumption that red-crowned cranes in Hokkaido are almost entirely dependent on human-supplied corn.”

Taking into consideration that this observation was reported on adult individuals, it may be an expected outcome, since cranes’ physiological need for animal-derived protein is higher during the breeding period.

- L 377-379: Add a paragraph with previous studies regarding the types of crops consumed by cranes, and also their high adaptability to forage on different kinds of agricultural crops depending on their availability in the landscape.

- Please discuss the factors affecting food choice by cranes.

- Add a comment regarding the fact that the digestive system of cranes is not adapted to digest high-fiber plant-foods, and cranes are dependent on invertebrate and vertebrate prey and plants with low fiber.

- Please discuss about the influence of agriculture on crane species: destruction of natural habitats and provision of cereal grain.

- Please add a comment sustained by cited references regarding the fact that the proportion of plants such as maize and rice grains in the diet of red-crowned cranes has increased, as a possible consequence of the degraded natural wetland habitat and the decreasing of the natural prey. This can partly be explained by supplementary food (i.e., maize) provided to red-crowned cranes as a compensatory measure at the degraded sites.

- L 413: Please add an estimation of the crop damage caused by red-crowned cranes (with a reference).

Conclusions 

The conclusions are not consistent with the stated results.

- L 435-438: This is not a conclusion of your study.

- L 438-441: These conclusions are not really justified based on the evidence presented, because results of this study showed that in southeastern Hokkaido red-crowned cranes did not cause significant damage to crops and significant economic loss for farmers, at least in terms of feeding damage.

- Please add some perspectives of this study.

Author Contributions:

- L 452-457: analysis from crane blood and muscles of cranes are not presented in this article!!

- L 456: data analysis is not presented. Please add this section.

Author Response

Abstract:

1) Please add numbers when you present your results.

Response: It is appreciated for many important comments. We have added the percentages for each result in Abstract, as highlighted in red in the revised manuscript. We are sorry, but the tomatoes in chicks were our mistake and have been removed.

2) L 34-35: Please rephrase “Additionally, there was little difference between the crop feeding status of cranes in winter and other seasons”. What does it mean “little difference”?

Response: The relevant text was inappropriate. We changed the sentence to “There was no significant difference in crop intake status in winter and that in other seasons for most of the crops”.

Introduction:

3) The introduction is incomplete. More comprehensive and recent literature review is needed.

Response: There are not many papers on the feeding status of cranes in Hokkaido, not only on crops consumption. On the other hand, there have been several recent reports on continental populations and other crane species. We have added some additional information on crops species with old and recent publications (L68-77 in the revised manuscript).

4) Please add previous studies about the diet of red-crowned cranes

Response: In response to Editor’s suggestion, we added crops species that were mentioned by Kobayashi et al. (2002) and Masatomi (2000) as the crop consumption by red-crowned crane in Hokkaido (L68-71).

5) L 57: The reference [4] is inaccurate. Please check and correct.

Response: We checked and corrected it. We found publisher’s name was a little bit different.

6) L 57: add a comment (and a reference) on the decision of the MOEJ to gradually reduce the supply of dent corn in order to disperse the cranes, which may change their feeding status and may influence their survival rate.

Response: We added the sentence and a reference as suggested (L62-64).

7) L 60-66: This is unnecessary information for this manuscript.

Response: We removed it except one sentence that shows that HTS-based data supports previous observation using binoculars (L66-68).

8) L 63: Abbreviations should be defined when they firstly appear. Such as “HTS-based metabarcoding”

Response: In response to Reviewer (7), the relevant text has been deleted. We show  High-throughput sequencing (HTS) on first appearance in the revised manuscript.

9) L 69: add a reference.

Response: We added a reference (L84).

10) L 83-86: add a reference.

Response: We added two references (L97).

11) L 89-92: please formulate a purpose of your research article.

Response: We modified a sentence from L85 in the original manuscript to formulate a purpose of our research (L103-105).

Materials and methods

12) L 96-97: please add for each area the geographical coordinates.

Response: We added the coordinates of the central city of each branch in the revised manuscript (L113-115). Additionally, we enlarged inset to show each area in Figure 1 and indicated location of the central city of each branch.

13) L 97: give some information related to the study area (climate, temperature, precipitation, vegetation).

Response: We added the information on climate, temperature, precipitation in the study area [L117-119]. Since vegetation is very complex and should use a large space in the revised manuscript, we cited an adequate reference [L120].

14) L 101: add details regarding the collection of feces and the content of the small intestine. Explain the division of samples taking into consideration the feeding period.

Response: We added descriptions on the collection of feces and intestinal content as well as the division of samples according to the feeding period (L121-123; L126-131).

15) L 102: please add some details for samples preparation.

Response: We added some descriptions on samples preparation (L137-140).

16) L 103: remove “by”

Response: We removed it (L136).

17) L 103: add city and country for Qiagen.

Response: We added it (L136-137).

18) L 170: The section regarding data analysis is missing. There is no description of statistical analysis.

Response: We added a section on data analysis in the end of Materials and Methods (L209-212) and description on it in some parts of Results section (L253-255, L283-285, L312-314).

19) Dairy cattle feces are mentioned in this study. Please add the details related to the collection of these samples and describe the purpose of cattle feces collection.

Response: We mentioned the collection of cattle feces with the purpose (L132-134). We knew that corn kernels were included in the cattle feces, but there was no description in scientific papers at least in Japan as far as we knew.

Results:

20) L 182: the sentence is repeated. Please remove it.

Response: We are very sorry. We removed the second sentence.

21) L 183-187: the paragraph should be moved to Materials and methods

Response: We moved the relevant sentence to Materials and methods [L120-123]. However, we left modified forms of some sentences to indicate case numbers for A and B in a range of figure legend, and to explain about Erimo sample as a figure legend, since we fear that it would be very difficult for the reader to understand without these sentences.

Discussion:

There is little in the way of insightful discussion in this section.

Response: In response to comments from Reviewers, we tried to add insightful considerations by responding to Reviewers’ comments very carefully. However, we believe that enough discussion was included that supports our main conclusion that red-crowned cranes in Hokkaido are highly dependent on dairy farmers for their food supply.

22) L 319-320: “No relevant seasonal or regional differences in intake content and frequency were observed”. No differences were observed based on a statistical analysis?

Response: We performed Pearson's chi-squared test to analyze seasonal differences in intake, which supports that there were no overall seasonal differences (L253-255). For some crops, however, detection rates differed more than twofold between summer and winter. For example, the detection rate of tomatoes in the non-feeding period is more than twice that in feeding period. Conversely, Chinese cabbage and soybeans tended to be more abundant in the feeding period. Lettuce or prickly lettuce also tended to be more abundant in summer. We also added these sentences in the revised manuscript (L255-259). We modified the relevant sentence a little (L377-378).

23) L 346: “Thus, 39.1% (9/23) of the individuals had no corn detected in the feeding period. This is in contrast to assumption that red-crowned cranes in Hokkaido are almost entirely dependent on human-supplied corn.” Taking into consideration that this observation was reported on adult individuals, it may be an expected outcome, since cranes’ physiological need for animal-derived protein is higher during the breeding period.

Response: As Reviewer noticed, the results were very surprising. Although it has been believed that most cranes entirely depended on dent corn at their feeding sites during the winter, no studies have actually confirmed this in fact. We modified the sentence in the introduction to " This is in contrast to the speculation that ....." (L406-407). 

We considered the following three points.

This is very surprising indeed. Identifying the causes of this problem is important but very difficult at this time. The following three points are worth mentioning.

1) We compared the present data with our previous study on the animal food intake of the same red-crowned cranes examined in this study (Kataoka et al., J Vet Med Sci. 84: 358-367. 2022.). Individual crane in which no cones were detected did not have a higher number of animal species detected than those in which cones were detected {Number of animal species detected: corn (-) 2.3±0.7 (n=12) vs corn (+) 8.6±0.7 (n=23)}. We would like to compare the amount of intake; however, the quantifiability of HTS using fecal samples is unreliable [Ando et al., 2020, 18 in the reference list]. 

2) It is possible that they may be eating some other crop or plant species in addition to the ones examined in this study. It is possible to examine the number of plant species ingested by performing HST using multiple universal primers. But, it would be very difficult to integrate data obtained with multiple primer sets. Another problem is that the quantifiability of HTS using fecal samples is not reliable (Ando et al., 2020). However, due to snow and freezing of the ground in winter, it is unlikely that there are many calorie-rich wild plants in southwestern Hokkaido during the winter season.

3) Recent global warming may have expanded the number of feeding areas available for winter use. Interviews with farmers and field surveys are needed. 

 Some of these considerations (especially the third) were summarized in the revised manuscript (L408-410).

24) L 377-379: Add a paragraph with previous studies regarding the types of crops consumed by cranes, and also their high adaptability to forage on different kinds of agricultural crops depending on their availability in the landscape.

Response: We added a paragraph on the types of crops consumed by cranes (L451-459). We also add a sentence on the their high adaptability, using a phrase suggested by the reviewer (L447-449).

25) Please discuss the factors affecting food choice by cranes.

26) Add a comment regarding the fact that the digestive system of cranes is not adapted to digest high-fiber plant-foods, and cranes are dependent on invertebrate and vertebrate prey and plants with low fiber.

Response: We have collectively addressed the points raised in 25), 26). We added a paragraph on the factors affecting food choice by red-crowned cranes including the property of their digestive system (L451-459).

27) Please discuss about the influence of agriculture on crane species: destruction of natural habitats and provision of cereal grain.

Response: We added a discussion with an emphasis on red-crowned cranes. (L460-475).

28) Please add a comment sustained by cited references regarding the fact that the proportion of plants such as maize and rice grains in the diet of red-crowned cranes has increased, as a possible consequence of the degraded natural wetland habitat and the decreasing of the natural prey. This can partly be explained by supplementary food (i.e., maize) provided to red-crowned cranes as a compensatory measure at the degraded sites.

Response: Thank you for the important suggestion. We do not have data on a detailed time series, but it is believed that red-crowned cranes in Hokkaido were foraging in wetlands more than 80 years ago, so cranes must have increased their dependence on corn (Rice is rarely cultivated in southeastern Hokkaido.). Our study suggests that cranes feed on most of the corn in compost piles rather than in fields, so we can expect further dependence on cattle farmers if wetlands continue to decline in the future. We added a sentence on this discussion (L468-473).

29) L 413: Please add an estimation of the crop damage caused by red-crowned cranes (with a reference).

Response: It is very difficult depending on a few data on it, since we suggested that cranes do not so much forage in the fields, but probably feed mostly on compost piles and human leftovers based on this study. However, Hokkaido Government reports that farmers suffer $4,000 in damage annually (L89-90). This includes dent corn, damage to cattle, beets, potatoes and wheat. Of these, the cow indicates that the crane startles the cow, causing injury and reduced milking output. As for beets and potatoes, the cranes rarely eat them, but rather invade and destroy the fields. The amount of damage caused by cranes eating field crops is expected to be much less (L513-515).

Conclusions:

30) The conclusions are not consistent with the stated results.

Response: In response to the reviewer's suggestion, we have changed the sentences as follows. We moved discussion part in Conclusions.

31) L 435-438: This is not a conclusion of your study.

Response: We moved these sentences to Discussion section (L525-530).

32) L 438-441: These conclusions are not really justified based on the evidence presented, because results of this study showed that in southeastern Hokkaido red-crowned cranes did not cause significant damage to crops and significant economic loss for farmers, at least in terms of feeding damage.

Response: In fact, red-crowned cranes already breed in the central area of Hokkaido, which is also an agricultural area such as rice farming from several years ago (L91-94). Central area of Hokkaido is famous for rice cultivation, which are very different from southeastern Hokkaido. We do not know whether red-crowned cranes do not feed on rice including growing buds or not at present.

33) Please add some perspectives of this study.

Response: We added two points as perspectives of this study (L541-545).

Author Contributions:

34) L 452-457: analysis from crane blood and muscles of cranes are not presented in this article!!

Response: We are very sorry. We deleted blood and muscles in the sentence. Instead, we added collection of chick feces.

35) L 456: data analysis is not presented. Please add this section.

Response: We added data analysis with a co-author (Dr. Akira Sawada (A.S.), additional co-author from original manuscript).

Round 2

Reviewer 2 Report

I have no comment.

It seems that all parts of the manuscript amended.

Author Response

Thank you very much for your careful review of our manuscript.

Reviewer 4 Report

The paper has been carefully revised by the authors and it only needs to undergo a few minor changes:

-L 142, 143-144: "were collected in 2006-2021 and kept in a freezer until intestinal contents were obtained", "were collected in June and July in the period from 2016 to 2018 143 and stored in a freezer until analysis." This kind of information should appear only in Materials & Methods section.

- L 207: please mention the soft used for statistical analysis (and producer).

Author Response

The paper has been carefully revised by the authors and it only needs to undergo a few minor changes:

Thank you very much for your careful review of our manuscript. We added minor modifications in response to your suggestions.

1) L142, 143-144: "were collected in 2006-2021 and kept in a freezer until intestinal contents were obtained", "were collected in June and July in the period from 2016 to 2018 143 and stored in a freezer until analysis." This kind of information should appear only in Materials & Methods section.

Response: We deleted the relevant phrases as suggested (highlighted in red).  

2) L207: please mention the soft used for statistical analysis (and producer).

Response:  We used R version 4.2.2 (R Core Team 2022) for statistical analysis. We have added this information to the Materials and Methods section and added a reference to the reference list. We corrected subsequent misalignment of literature numbers in the second revised manuscript.  

R Core Team (2022). R: A language and environment for statistical computing. R Foundation for Statistical Computing, Vienna, Austria. URL https://www.R-project.org/.